# Cultural Tourism Weakens Seasonality: Empirical Analysis of Chinese Tourism Cities

**Jing Zhang [1], Zhonglei Yu [1,2], Changhong Miao [1,2,*], Yuting Li [1] and Shuai Qiao [1]**

[1] Key Research Institute of Yellow River Civilization and Sustainable Development & Collaborative Innovation Center on Yellow River Civilization Jointly Built by Henan Province and Ministry of Education, Henan University, Kaifeng 475001, China; 104753181181@vip.henu.edu.cn (J.Z.); yzlei@mail.bnu.edu.cn (Z.Y.); liyuting@henu.edu.cn (Y.L.); 104753201432@henu.edu.cn (S.Q.)

[2] College of Geography and Environmental Science, Henan University, Kaifeng 475004, China

[*] Correspondence: chhmiao@henu.edu.cn

**Abstract:** Cultural tourism is less seasonal than nature tourism. However, previous studies have mainly compared the tourist flow of scenic spots, and it is still unclear how cultural tourism affects regional tourism seasonality. This study investigated the seasonal patterns by analyzing the monthly inbound tourist flow of the 28 typical Chinese tourist cities from 2001 to 2012, and examined the effect of cultural tourism on weakening seasonality by using the random-response panel Tobit model. It was found that the seasonal patterns of inbound tourists present non-peak, one-peak, two-double, and three-peak regular fluctuations, and also have irregular fluctuations affected by emergencies and festivals. Cultural tourism can weaken the seasonality of regional tourism, while nature tourism products enhance tourism seasonality. Suitable travel times throughout the year and destination reception industry levels, locations, and external connections have a significant impact on the regional tourism seasonality, but climate comfort, foreign direct investment, and simply the number of hotels and international travel agencies are not significant for tourism seasonality.

**Keywords:** cultural tourism; regional tourism seasonality; inbound tourist; Tobit mode

## 1. Introduction

Seasonality is a globally recognized basic characteristic and challenge of tourism, which affects the sustainable development of tourism [1,2]. Almost all world destinations are facing a seasonal concentration of tourist activities. The destination suffers from low resource utilization efficiency, near-fracture of the supply chain, and unemployment of employees in off-season tourism. The tourists suffer from price increases and service quality declines in the peak season, resulting in lower satisfaction [3,4]. Additionally, the excessive concentration of tourist flow can easily cause environmental problems such as ecological degradation, garbage flooding, water supply issues, and waste management issues [5,6]. The local residents' well-being also declines due to traffic congestion and rising living costs [7]. Therefore, tourism seasonality has become an important issue of great concern to scholars, governments, tourism managers, and other sectors of society.

Scholars have conducted fruitful research on the seasonality, mainly including definitions, patterns, causes, implications, strategies, and conclusions. A regularly cited definition was provided by Butler who defined the concept of tourism seasonality as the temporary imbalance in the phenomenon of tourism, which may be expressed in terms of the number of visitors, traffic on the highways, and employment and admission to attractions. Seasonality is generally characterized by relatively stable patterns, rather than random irregularities [8]. According to the temporal agglomeration characteristics of tourist flow, seasonality can be divided into non-peak, one-peak, and two(multi)-peak [9,10]. In Spain, as with Portugal, Greece and Cyprus, the one-peak pattern is predominant, mainly in sun and sand destinations, and the peak season tends to be in summer [11]. However, China is

different. China's inbound tourism peak seasons are concentrated in March, April, July, August, October, and December [12]. The seasonal patterns of coastal and inland cities, tourist cities of different latitudes, and different types of tourism products (i.e., ice and snow; seaside; mountains; cultural) are quite different [12,13]. Seasonal patterns in China are more complex and diverse, but there are few existing international studies.

Regarding the determinants of seasonality, scholars have focused mainly on natural (i.e., temperature, sunlight, and rainfall) and institutional factors (i.e., the schedule of school holidays; the planning and scheduling of festivals or cultural events in tourism destinations) [4,5,8,10]. However, other studies have emphasized the link between seasonality and the variety of the tourist product offered by the destination. In addition, the characteristics of tourists' age [14,15] and tourism motivation [11,14], and social trends [16] are important drivers of seasonality. Although these variables have a profound impact on tourist decision-making, for a specific tourism destination, the nature [17–19], location [1,12,19], reception level [17,20,21], special events [22,23], tourism resources, and destination types [13,24,25] are the dominant factors shaping the distribution pattern of tourist flow in the time series.

Scholars believe that the determinants of seasonality are different. Early studies believed that natural factors determine the seasonality of tourist destinations [18,26]. With the continuous development of empirical research, most scholars believe that socio-economic, cultural, and natural factors jointly determine the tourism seasonality [27]. However, with tourism development, tourism resources are developed in different seasons, tourism activities are increasingly abundant, and tourism motivation tends to be diversified. The socio-economic and cultural factors on tourism seasonality is more significant [28]. For example, empirical studies have shown that high-quality tourism services and convenient transportation locations weaken the seasonality [12,17,19–21], especially comprehensive metropolises as tourism distribution centers [1,8,12]. Over the past decade, many international scholars have begun to compare and analyze the seasonality of cultural, natural, and comprehensive scenic spots, and to find that the seasonality of cultural tourism is relatively weak [11,20,25,29,30]. Therefore, they believed that the development of cultural tourism resources could avoid strong seasonality. There are similar studies in China. Zhong compared different historical and cultural villages and towns [13], and Yu compared Huangshan, Xidi, and Hongcun [24]. They also found that cultural tourism destinations could weaken seasonality. However, based on the empirical research of Spain, Saito believed that the number of museums and natural factors contribute similarly to regional seasonality [31].

Seasonality has diverse and complex impacts. Although many authors have considered that seasonality has numerous negative repercussions for economy, employment, the environment, and society, other researchers have paid attention to its potential benefits [1,4,5]. For instance, during the off-season, ecological and socio-cultural recovery takes place, as well as maintenance and reform of tourist infrastructures [4–6]. In addition, in periods of greater demand, temporary workers (i.e., students and artists) can be incorporated into labor markets. In general, the negative effects of seasonality are greater [1,4,5].

Thus, there are many strategies that are used to address the effects of seasonality, at both the enterprise and destination. Differential pricing, diversified attraction (i.e., festivals and events), market diversification (tourists), taxes, etc. are recognized strategies to reduce tourism seasonality [1–5]. Moreover, mature tourism destinations, improving the convenience of national transportation [4,27], and developing cultural tourism [11,20] are also the keys to reducing seasonality.

Whether cultural tourism can weaken the seasonality has not yet reached a consensus. This needs more empirical research not only on scenic spots but also on the regional level. First, regions are an important spatial unit of tourism activities. A region often contains different types of tourism attractions, and the tourist flow of cultural tourism scenic spots is also easily affected by adjacent scenic spots. The seasonal weakening effect of cultural tourism can be more accurately reflected on the regional scale. Secondly, from the perspective of evolution, the types of destination tourism products will change with the development of tourism resources [29,32]. Research on seasonal patterns over a long

period at the regional level can more effectively identify the seasonal weakening effect of cultural tourism. Therefore, this study aimed to reveal whether the development of cultural tourism can weaken the seasonality at the regional level based on some typical tourist cities in China.

## 2. Materials and Methods

### 2.1. Research Object and Data Source

This study focused on 28 tourist cities in China (Beijing, Tianjin, Shenyang, Dalian, Changchun, Harbin, Shanghai, Nanjing, Wuxi, Suzhou, Hangzhou, Ningbo, Huangshan, Xiamen, Jinan, Qingdao, Wuhan, Guangzhou, Shenzhen, Zhuhai, Zhongshan, Guilin, Haikou, Sanya, Chongqing, Chengdu, Kunming, and Xi'an), including international tourist distribution centers, provincial capital cities, port cities, and other types of tourism cities that are representative. The research period was from 2002 to 2012. As Severe Acute Respiratory Syndrome (SARS) caused an unexpected disturbance to the inbound tourist flow in 2003, this year was not included in the analysis. As only a few cities publish the monthly tourist flow of domestic tourism, and it is difficult to unify the year, the monthly inbound tourist flow data are more comprehensive, and the statistical caliber is more reliable. This study mainly investigated the seasonality of inbound tourism. The monthly inbound tourist flow came from the information network of the development research center of the State Council and the China Tourism Statistics Yearbook (https://www.drc.gov.cn/default.aspx; https://data.cnki.net/yearbook. Both link were last accessed on 8 August 2020). Other data in this paper included (a) the number of international travel agencies and star hotels derived from China's Tourism Statistical Yearbook, and some missing data supplemented by the average growth rate and interpolation method; (b) meteorological data such as temperature and humidity used to calculate climate comfort and annual suitable tourism duration derived from China's meteorological data (http://data.cma.cn. The link was last accessed on 16 July 2020); (c) 4A and 5A scenic spots, world heritage sites, and famous historical and cultural cities derived from the official websites of the Municipal Bureau of culture and tourism; (d) the distance between each city and Beijing, Shanghai, and Guangzhou entry port cities being the shortest road distance between cities, which can be obtained by querying the Baidu map (query time is July 2020).

### 2.2. Seasonal Measurement Method

Common measurement methods of seasonality include the seasonal intensity index, Gini index, Theil index, seasonal range, seasonal ratio, seasonal variation index, and entropy method [9,33]. The seasonal range and seasonal ratio mainly measure the seasonal fluctuation range, which is vulnerable to the maximum and extreme values. The Gini index, Theil index, and seasonal variation index eliminate dimensions and reflect the degree of seasonal variation [34]. The seasonal intensity index, Gini index, and Theil index are highly alternative [2]. The seasonality measured by these three methods often has high consistency [2,17,34]. The Gini index has a high utilization rate at home and abroad because of its relative stability, low sensitivity to extreme values, and dependence on monthly peak changes. Accordingly, the Gini index was used to measure the seasonality of 28 typical tourist cities in China on the two timescales of the month and the quarter. The Gini index is calculated as follows:

$$G = \frac{2}{n}\left[\sum_{i=1}^{n} if_i - \frac{n+2}{2}\right] \quad (1)$$

G is the Gini index, usually between 0 and 1. The larger the Gini index, the more concentrated the distribution of tourist flow in time; N is the total number of months in a year, n = 12; i represents the number of inbound tourists received each month in a year in ascending order; $f_i$ is the ratio of the number of tourists received in month i to the total number of tourists received in the year.

### 2.3. Measurement Model Construction

2.3.1. Variable Selection

This study focused on whether cultural tourism helps to weaken the tourism seasonality of the destination. However, considering that tourism seasonality results from the comprehensive action of multiple factors, referring to existing studies [19,24,35–39], climate, location, foreign economic relations, and the development level of the tourism reception industry were included as control variables (Table 1).

**Table 1.** Independent variables of tourism seasonal factors.

| Title 1 | Variable Name | Mean | Standard Deviation | Expected Symbol |
|---|---|---|---|---|
| Tourism product structure | Cultural tourism attractions | 38.7240 | 44.7379 | − |
| | Nature tourist attraction | 24.4058 | 24.1987 | + |
| | Comprehensive tourist attraction | 3.8246 | 6.9306 | +/− |
| Climatic condition | Climatic comfort | 5.4973 | 0.5676 | − |
| | Suitable length of travel throughout the year | 152.1429 | 29.9721 | − |
| Location and external economic connection | Whether it is the first two batches of coastal open cities * | 0.2857 | 0.4524 | − |
| | Distance from international distribution center | 576.5929 | 520.1601 | + |
| | Foreign direct investment | 4.4146 | 1.3993 | − |
| The development level of the destination's international hospitality industry | Number of star hotels | 126.5455 | 118.9567 | +/− |
| | Number of international travel agencies | 31.1591 | 36.4102 | +/− |
| | Hotel and travel agency cross term | 7.7265 | 1.2422 | − |

* Virtual variable.

(a) Explanatory variables

Regions are an important tourism destination, the carrier of tourism resource development, and the center of tourism distribution and activities. Cultural tourism is regarded as a "good medicine" to avoid seasonality. On the one hand, because most of the places where cultural tourism is carried out, such as cultural sites, theme parks, and museums, are indoor places and are less affected by the climate, tourists' tourism motivation and cultural experience will not decrease with climate change. On the other hand, it is generally believed that cultural tourists are more aged than noncultural tourists with higher education and socioeconomic status [15]. Older tourists are not affected by the holiday system. In principle, they can travel freely year-round. The regions dominated by nature tourism resources are more affected by climate factors and attract younger tourists, so they have strong seasonality. In this study, the number and level of natural, cultural, and comprehensive scenic spots were weighted to represent the development level of different types of tourism products.

Inbound tourists have a wide range of activities, short travel time, and high disposable income, so tend to visit scenic spots or regions with high levels. This study referred to the existing literature to empower the double world heritage, world cultural heritage, world natural heritage, famous historical and cultural cities, 4A and 5A scenic spots, famous historical and cultural cities, and national geo-parks [19,40]. They were, respectively, assigned as 13, 10, 8, 6, and 5 (there was no repeated scoring in the measurement process, and they were updated and recalculated every year). According to the classification standard of tourism resource attributes, they were divided into three types: natural type, cultural type, and comprehensive type, and their scores were summarized and calculated.

(b) Control variable

Climatic factors. Climate is an external environmental factor supporting tourism activities and an important tourism attraction [41]. The comfort and duration of the

regional climate are important factors affecting the choice of tourist destination and the seasonal change of tourism [42]. However, a favorable climatic factor could be a necessary but not sufficient condition to avoid seasonality in tourism. Climate comfort refers to the comfortable range in which people can ensure the normal activities of physiological processes without the help of any cold and summer measures. Some classical climate comfort calculations (and their improved forms), such as the temperature and humidity index and wind efficiency index, have become the measurement indicators often used in human comfort prediction [43,44] in meteorology and climate comfort evaluation in geography [45–50]. This study used the temperature and humidity index (THI), wind chill index (WCI), and clothing index (ICL) to express the regional tourism climate comfort. For the specific calculation formula, they were unified into a unified index by the weighting method [38]. The calculation formula is:

$$C = 0.6X_{THI} + 0.3X_{WCI} + 0.1X_{ICL} \tag{2}$$

$X_{thi}$, $X_{wci}$, and $X_{icl}$ are the graded assignments of the temperature-humidity index, wind-cold index, and clothing index, respectively, and 0.6, 0.3, and 0.1 are the weight coefficients of each subindex. When $7 \leq C \leq 9$, it is considered very comfortable. When $5 \leq C < 7$, it is considered comfortable. When $3 < C < 5$, it is considered less comfortable, and when $1 < C \leq 3$, it is considered uncomfortable. The suitable travel time is the sum of the number of months with a monthly climate comfort of $5 \leq C \leq 9$.

Location and external economic relations: First, the accessibility of the destination is one of the influencing factors of the spatial difference of inbound tourist flow, and it is also one of the reasons for seasonality [35]. Convenient traffic locations can mitigate the seasonal fluctuation of tourism [17,21,27]. Inbound tourism mostly enters tourist destinations through an international or regional distribution center [1,8]. There are many rapid transportation modes (i.e., aviation, high-speed rail, and highway) in Beijing, Shanghai, and Guangzhou, making it the most advantageous in distribution [51,52]. Most of the other regions transit through these three regions. Thus, the general itinerary of inbound tourist flow is "International Distribution Center—regional distribution center/tourism destination". In addition, inbound tourism has a spatial distance attenuation effect [53]. Tourism demand reaches its peak at a spatial distance close to the market and then decreases exponentially with distance. The actual spatial distance and perceived distance between the source and destination affect the tourist decision. Therefore, this study chose the distance from the Beijing, Shanghai, and Guangzhou international distribution centers as control variables. Secondly, due to China's special national conditions, it was the first to become a coastal open port city, such as Dalian, Qingdao, Ningbo, and Shanghai. They were the first to become a trading port in history, and there are many historical buildings. International business travel is frequent in these cities because weather conditions do not limit business travel. Destinations solve the problem of off-season tourism demand by holding business and tourism activities (i.e., international conferences, international exhibitions, and international events) [31]. In addition, these regions developed tourism early and have strong accessibility (i.e., language and culture, transportation, visa system, and friendliness), and then further improve the strength and popularity through the cumulative effect of tourism. Therefore, this study selected the first two batches of open port cities as control variables. Thirdly, foreign direct investment has a close relationship with inbound business tourism, and the distribution of foreign direct investment is roughly the same as that of inbound business tourists [38]. Business tourism can weaken the seasonality of tourism through business activities, so it is considered that foreign direct investment is one of the factors affecting the seasonal fluctuation of inbound tourists. Foreign travel agencies also tend to consume star hotels and restaurants established by foreign investment when arranging outbound tourism, so this study selected foreign direct investment as one of the control variables.

The reception level of destination tourism: The development of tourism activities is based on the local accommodation industry and catering industry. Star hotels are the

material conditions for inbound tourism activities [40], and high-quality hotels have longer peak tourism seasons and weak tourism seasonality [21,45]. At the same time, inbound tourists are far from the destination city's spatial distance and cultural perception, and they basically rely on international travel agencies to organize inbound and outbound activities. International travel agencies have become an important factor restricting regional tourism development [38]. This study selected the number of star hotels and international travel agencies as control variables. Considering the causal relationship between the level of the reception industry and the amount of inbound tourism reception, two periods in advance can avoid endogeneity. Due to the high correlation between star hotels and international travel agencies, in order to avoid collinearity, this study also chose the intersection of star hotels and international travel agencies two periods in advance to measure the reception level of destination inbound tourism.

### 2.3.2. Model Building

This study used the Gini index to measure tourism seasonality, and its value range was [0, 1]. It has the characteristics of segmentation and meets the setting conditions of the restricted-dependent-variable Tobit regression model [54]. Moreover, compared with the fixed-effect panel Tobit model, the random-effect panel Tobit model can obtain consistent estimation [55] and can effectively avoid the bias of the results brought by the least squares method [56]. Therefore, the random-effect panel Tobit model was adopted in this study. The mathematical expression of the model is as follows:

$$\begin{aligned} Gini_{it} = {} & \beta_0 + \beta_1 C_{it} + \beta_2 Time_{it} + \beta_3 Cul_{it} + \beta_4 Nat_{it} + \beta_5 Gen_{it} + \beta_6 lnfdi_{it} \\ & + \beta_7 Port_i + \beta_8 Dis_{it} + \beta_9 Hotel_{i,t-2} + \beta_{10} Agen_{i,t-2} + \beta_{11} lnSer_{i,t-2} + \varepsilon_{it} \end{aligned} \tag{3}$$

$Gin_{it}$ is the Gini index, i is the city, and t is the time. C is the annual average climate comfort. Time is the annual suitable tourism duration. Cul is the weighted score of cultural tourism resources. Nat is the weighted score of nature tourism resources. Gen is the weighted score of comprehensive tourism resources. lnfdi is foreign direct investment. Port is the dummy variable, whether it is the first two batches of coastal cities open to the outside world. Dis is the distance from Beijing, Shanghai, and Guangzhou. Hotel is the number of star hotels in advance. Agen is the number of international travel agencies in advance. lnSer is the cross term of star hotels and international travel agencies in advance. The reception level of tourism and cons is a constant term. Using Stata 15.1 metrological analysis software, the correlation coefficients between the number of star hotels and the number of international travel agencies in the model is high (0.871).

## 3. Results Analysis and Discussion

### 3.1. Seasonal Characteristics of Tourism

From the time agglomeration characteristics of inbound tourist flow, the tourism seasonality of 28 cities in this study mainly presents five patterns: non-peak, one-peak, two-peak, three-peak, and irregular fluctuations.

- Non-peak: Tianjin, Dalian, Shanghai, Ningbo Qingdao (2001–2007), and Suzhou are non-peak types (Figure 1). Suzhou is a pure cultural tourism city, and the annual inbound tourist flow is the most stable. The inbound tourist flow curves of the top five cities except for Suzhou from 2001 to 2012 (excluding 2003) are relatively stable and only slightly decrease in January and February of each year. These cities were the first or second batch of coastal open port cities after the reform and opening up. Their foreign trade is prosperous, and inbound tourism is opened earlier, which greatly impacts the tourism seasonality.
- One-peak: Xiamen, Shenzhen, Zhuhai, and Zhongshan are one-peak, and the number of inbound tourists in December of each year is higher, but the other months are relatively stable (Figure 2). These four cities are well-known hometowns of overseas Chinese. Every December is the Christmas holiday abroad, and the tide of over-

seas Chinese returning home to visit their relatives has caused a surge in inbound tourist flow.

- Two-peak: Beijing, Nanjing, Wuxi, Shenyang, Changchun, Harbin, Wuhan, and Xi'an are two-peak. The peaks of Beijing, Nanjing, and Wuxi are from March to May and from September to November each year (Figure 3). The peaks of Shenyang, Changchun, and Harbin (Figure 3) are all located in the northeast and are relatively consistent from May to July and November to January of the next year. The increase in tourist flow in February and March may lead to a change in the peak state due to ice and snow tourism development. The peaks of Wuhan and Xi'an are from March to June and September to November (Figure 3). The two cities are located in the central and western regions, with less tourist flow in July and August, which may be affected by the hot climate.

- Three-peak: Haikou, Sanya, and Hangzhou Jinan are three-peak (Figure 4). The peaks of Haikou and Sanya Island cities are from March to April, from June to August, and from November to January of the next year, which are different from the May Day golden week and the 11th Golden Week holidays in China. It may be that the increase in tourism costs makes inbound tourists choose off-peak travel. The peaks of Hangzhou and Jinan are from March to April, from June to July, and from September to November. May is the domestic golden week, while the hot climate in August in the two cities in May is one of the possible reasons for the trough. Additionally, although the two cities have profound historical and cultural buildings, there are few cultural heritages on the land surface, inbound tourists prefer world natural and cultural heritage sites, and cities with a low taste of tourism resources cannot benefit from the peak season [19]. In general, the above seasonal fluctuations of tourism are regular and periodic.

- Irregular fluctuations: Some cities show irregular fluctuations in special years [28]. First, affected by SARS in 2003, the inbound tourist flow in all cities changed greatly, resulting in great changes in characteristics. Affected by the Wenchuan earthquake in 2008, the inbound tourist flow in Chengdu fell in May 2008 and did not return to normal until 2010. Dalian, Qingdao, and other cities had great changes in 2008, and their foreign trade was prosperous, so they were strongly affected by the global financial crisis. The sudden drop in inbound tourist flow in Tianjin in July 2012 may be due to the explosion of local chemical plants and the environmental pollution caused. These seasonal fluctuations are the occurrence of emergencies, and their impact is violent, long-term, and with slow recovery. Therefore, it is necessary to enhance the resilience of regional tourism under emergencies. Second, in August 2008, there was a small peak in Beijing, from a two-peak to three-peak, but the peak returned to two-peak in 2009. Hosting the Beijing Olympic Games increased but did not have a lasting effect on inbound tourists. Compared with other years, the inbound tourist flow in Shanghai also increased significantly from May to July 2010, but the tourist flow in September and October was low. The holding of Shanghai World Expo increased the inbound tourist flow in a short time, but the inbound tourist flow a few months after "overdrafted"; November was originally the off-season for tourism in Guilin, but there was a small peak in November 2011, which may be affected by the Guilin International Wetland Culture Festival.

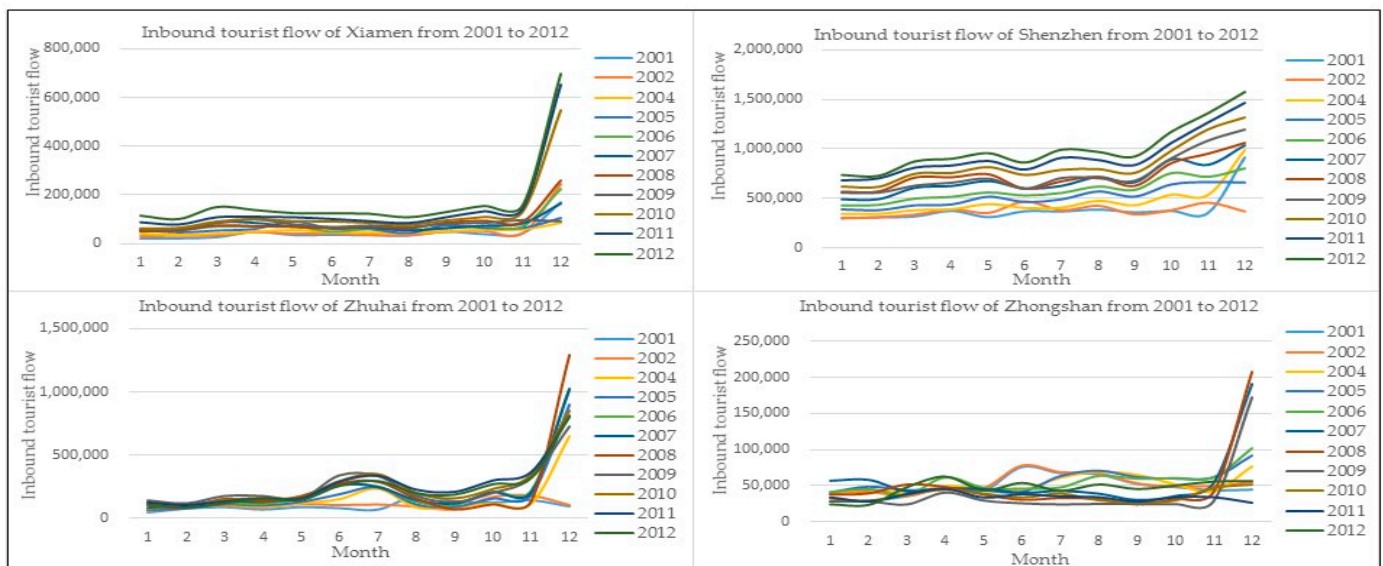

**Figure 1.** Non-peak. Source: authors' research and edited, and same as Figures 2–4.

**Figure 2.** One-peak.

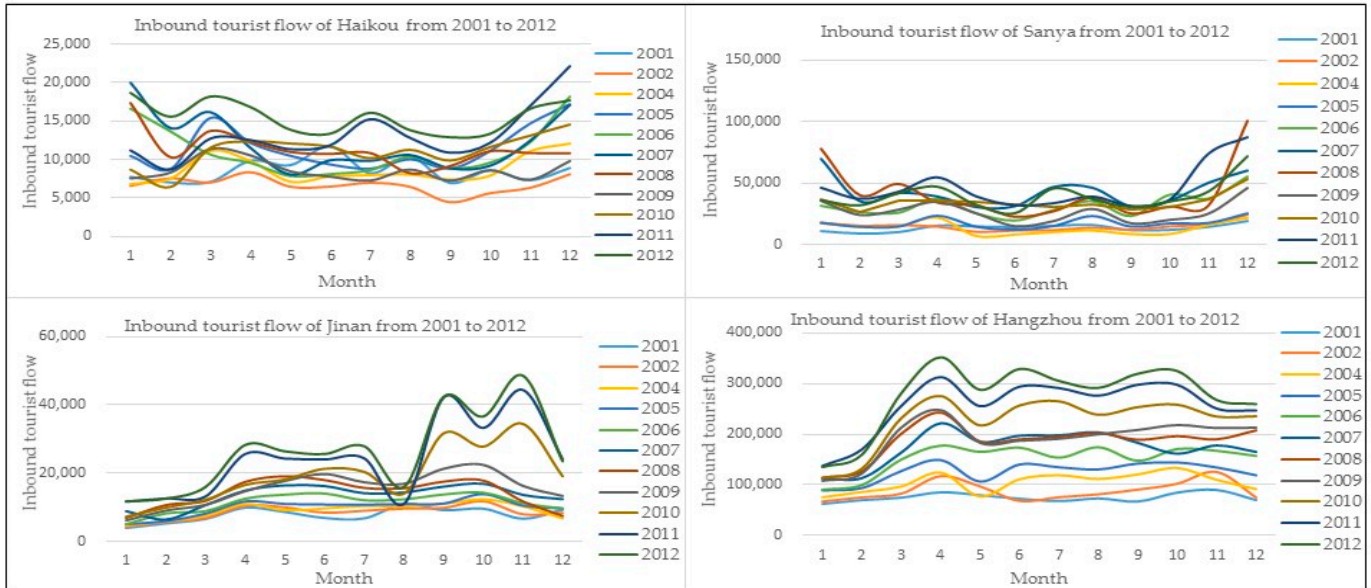

**Figure 3.** Two-peak.

**Figure 4.** Three-peak.

In summary, festival activities significantly impact the seasonal peak of tourism and bring concentrated and short-term fluctuations in tourist flow. Therefore, we should use the positive role of international exhibitions, sports events, and other festival activities in increasing tourist flow. At the same time, it should also be noted that the impact of festival activities on tourist flow is temporary, does not change the seasonal structure of destination [55], and may "overdraft" the subsequent tourist flow.

### 3.2. Regression Result Analysis

This study used the panel Tobit model to quantitatively identify the influencing factors of inbound tourism seasonality (monthly) in 28 typical tourism cities. The results show that the tourism product structure, climate conditions, location, external relations, and tourism reception industry significantly impact seasonality (Table 2). To test the robustness of the regression model, this study further used the Gini index of inbound tourist flow on a quarterly scale as the dependent variable to estimate the Tobit panel model. The results are basically consistent with the monthly seasonal regression model, and the significance level of only some explanatory variables changes from 1% to 5%. This shows that the estimation model in this study is robust, which is based on the monthly seasonal regression results that are analyzed and discussed.

**Table 2.** Regression results of factors influencing seasonal changes on a monthly and quarterly basis.

| Variable Name | Monthly Gini Coefficient | | Quarterly Gini Coefficient | |
|---|---|---|---|---|
| | Coefficient | $p$ Value [1] | Coefficient | $p$ Value [1] |
| Cultural tourism attractions | −0.00049 | 0.000 *** | −0.00028 | 0.021 ** |
| Nature tourist attraction | 0.00053 | 0.017 ** | 0.00051 | 0.009 *** |
| Comprehensive tourist attraction | 0.00104 | 0.352 | −0.00009 | 0.924 |
| Climatic comfort | 0.117 | 0.205 | 0.00992 | 0.216 |
| Suitable length of travel throughout the year | −0.00052 | 0.006 *** | −0.00030 | 0.063 ** |
| Whether it is the first two batches of coastal open cities | −0.03853 | 0.001 *** | −0.02159 | 0.036 ** |
| Distance from international distribution center | 0.00003 | 0.000 *** | 0.00003 | 0.001 *** |
| Foreign direct investment | 0.00672 | 0.121 | 0.00375 | 0.317 |
| Number of star hotels | −0.00001 | 0.922 | −0.00006 | 0.556 |
| Number of international travel agencies | 0.00024 | 0.432 | 0.00030 | 0.246 |
| Hotel and travel agency cross term | −0.02154 | 0.003 *** | −0.01272 | 0.043 ** |
| Cons | 0.30570 | 0.000 *** | 0.18444 | 0.000 *** |

[1] $p$ Value is significance level. *** represents $p < 0.01$, ** means $p < 0.05$, * represents $p < 0.1$.

In terms of tourism product structure (Table 2), the regression results show that the regression coefficients of cultural, natural, and comprehensive tourism products on seasonal effects are −0.0005 ($p < 0.01$), 0.0005 ($p < 0.05$), 0.00104, respectively. This verifies the research hypothesis of this study and the views of some existing literature. Cultural tourism has a significant positive impact on reducing tourism seasonality at the regional level, while regions dominated by nature tourism products are vulnerable to climate [32]. The possible reasons are as emphasized in previous studies. First, most places for cultural tourism, such as cultural sites, theme parks, and museums, are indoor spaces, which are less affected by the climate [11]. Regardless of tourists' motivation to go to the destination, they often visit typical cultural tourism scenic spots, making the seasonality of cultural tourism-leading destinations weak [11]. Secondly, compared with the whole tourism group, cultural tourists have higher education, economic, and social status, are older, and are less affected by the vacation system, so their seasonality is weak [41]. Thirdly, the development of tourist products will change the tourism seasonality pattern of destinations. Compared with nature tourism products, cultural tourism products are easier to develop, thus weakening the tourism seasonality.

In terms of climatic conditions, the regression coefficient of suitable travel time in the whole year is −0.0005 (*p* < 1%), and the climate comfort is not significant (Table 2). This shows that the longer the suitable travel time throughout the year, the weaker the tourism seasonality, but the overall climate comfort of the destination has no significant impact on the seasonal fluctuation. The reason is that the longer the suitable travel time of the tourism destination, the larger the time range of choosing the destination, and the lower the time concentration of tourist flow. Travel agencies are also limited by the cost and popular routes when organizing inbound tourism activities, so there are more alternative periods of peak staggering for promotion. At the same time, previous studies have only shown that the higher the monthly climate comfort is, the more tourist flow there is, but the climate only partially affects tourists' decision-making [28]. The climate has a periodic impact on tourism seasonality, which is no longer the main factor of seasonality of cultural destinations [25].

In terms of location conditions and external relations, the influence coefficient of the first two batches of coastal open cities, the distance coefficient from Beijing, Shanghai, and Guangzhou international port cities on tourism seasonality is −0.0385 (*p* < 1%), 0.00003 (*p* < 1%), respectively (Table 2), which shows that external relations and location factors have a significant impact on the seasonality of tourism. In modern history, the first two groups of cities that opened to the outside world were mostly trade ports, with close trade and exchanges with foreign countries, strong cultural identity, a good foundation for tourism development, and more business tourism making the monthly inbound tourist flow more stable. In other words, China's inbound tourist flow mainly enters through major international ports such as Beijing, Shanghai, and Guangzhou, and tourists are often affected by spatial distance when choosing their destination. The closer the space between the tourist destination and major international port cities (such as airports or other transportation hubs) is to the entry port, the higher the probability of destination selection, and the weaker the fluctuation of inbound tourist flow, which also confirms the view of previous studies [11]. Contrary to the expectation, the impact of foreign direct investment on the seasonality is not significant. The reason is that although the spatial distribution of foreign investment is consistent with that of inbound tourist flow, it only shows that there are more inbound tourists in areas with more foreign investment. However, the temporal distribution of inbound tourist flow is also affected by other factors, which is not significant to the seasonal fluctuation of tourism.

Regarding the development level of the international reception industry in tourism destinations, the regression results show that the number of star hotels and international travel agencies are not significant. However, the cross-term coefficient is −0.02154 (*p* < 1%, Table 2), which shows that a high-quality and sufficient international tourism reception industry can weaken the seasonality of tourist destinations. The reason is that the development level of star hotels is affected not only by inbound tourism but also by the economic development level of the destination; the number of international travel agencies is affected not only by inbound tourism but also by local outbound tourism. The cross term of star hotels and international travel agencies can effectively represent the development level of the inbound tourism reception industry, which is significantly negatively correlated with the seasonality of tourism destinations, indicating that the high development level of the tourism destination reception industry helps to weaken the seasonality of tourism. The possible reason is that the tourism reception industry is an important part of destination tourism. A high-quality and abundant reception industry helps enhance the attraction of destinations and expand the distribution period of tourist flow.

## 4. Conclusions and Discussion

Seasonality is one of the most remarkable characteristics and challenges of tourism. Although cultural tourism is considered a key resource to counteract seasonality, more empirical research at the regional level is necessary. Based on the monthly inbound tourist flows of typical tourist cities in China from 2001 to 2012, this study analyzed the seasonal

patterns and examined whether cultural tourism can weaken seasonality at the region level. First, the seasonal patterns in China are diverse, showing five patterns: non-peak, one-peak, tow-peak, three-peak, and irregular fluctuations due to the impact of emergencies and festivals. Notably, cultural tourism destinations have weaker seasonal fluctuations. Regardless of the model, the seasonal fluctuations of cities dominated by cultural tourism products are smaller than those of nature tourism products. Unexpectedly, unlike the countries along the Mediterranean coast, the port cities along the coast of China are less seasonal. Overseas Chinese cities are usually one-peak, with the peak in December. Most of the seasonal patterns in eastern non-port cities are two-peak, and China's central and western mainland and island-type tourist cities are three-peak. Furthermore, the same conclusion as the existing research is that emergencies will cause a sudden drop in tourist flow and have a strong impact to a large degree, a long duration, and a slow recovery, and festival activities will promote a small tourist flow peak, but its impact on seasonality is concentrated, short-lived, recovers quickly, and may "overdraft" the tourist flow in subsequent periods. Secondly, tourism seasonality is affected by multiple factors, such as destination tourism product type, climate, location, external contact, and reception level. Cultural tourism has a weakening effect on regional seasonality, which is obvious and results from the essence of this kind of tourism.

This study confirms that cultural tourism can weaken seasonality at the regional level, breaking through the research at the scenic spot level in China, and has implications for the marketing and policy making of tourism destinations. First, it is necessary to rationally configure the type of structure of tourism products in regional tourism planning. For regions dominated by nature tourism attractions, we should pay attention to the development of cultural tourism products to reduce the seasonality of regional tourism. Secondly, we should actively hold various festivals and events to attract tourist flow and improve the development level of the tourism reception industry to reduce the seasonality of tourism. Thirdly, we should pay full attention to the location gradient effect of tourism seasonality. Regions far away from the tourism distribution center should strengthen destination marketing, improve travel convenience, and reduce travel costs to reduce the seasonality caused by spatial distance.

This study still had some limitations. Due to limitation of regional data availability, the time period for this study was 2001 to 2012. The patterns of inbound tourist flow in Chinese tourism cities from 2001 to 2012 showed that the number of tourists in most cities had grown steadily during the periods, and the seasonal fluctuations were also consistent, which indicate that it had a strong regularity. The seasonal fluctuation patterns had been affected by emergencies, festivals, tourism product development, and other factors, but the pattern of irregular fluctuations could return to regular fluctuations after a period of time, so it is very important for China's tourism today and still has modern significance. In particular, research on seasonality and the influencing factors of domestic and inbound tourist flow by choosing more cities in recent years will provide a significant contribution to understand the tourism resilience under the current impact of the COVID-19 pandemic.

**Author Contributions:** Conceptualization, J.Z., Z.Y. and C.M.; methodology, J.Z., Z.Y. and C.M.; software, J.Z.; validation, J.Z. and Z.Y.; formal analysis, J.Z. and Z.Y.; resources, J.Z.; data curation, J.Z. and S.Q.; writing—original draft preparation, J.Z., Z.Y. and C.M.; writing—review and editing, J.Z., Z.Y. and C.M.; visualization, Y.L. and S.Q.; supervision, Y.L. and S.Q.; project administration, Y.L. and S.Q.; funding acquisition, C.M. and Z.Y. All authors have read and agreed to the published version of the manuscript.

**Funding:** This research was funded by the National Natural Science Foundation of China (Fund project host: Miao Changhong), grant number 42171186; The Major Project of China National Social Science Fund in Art (Fund sub-project host: Miao Changhong), grant number 21ZD03; The National Natural Science Foundation of China, grant number 41901206 (Fund project host: Yu Zhonglei); Philosophy and Social Science planning project of Henan Province (Fund project host: Yu Zhonglei), grant number 2020CSH029.

**Institutional Review Board Statement:** Not applicable.

**Informed Consent Statement:** Informed consent was obtained from all subjects involved in the study.

**Data Availability Statement:** The monthly inbound tourist flow reference data: the information network of the development research center of the State Council and the China Tourism Statistics Yearbook, https://www.drc.gov.cn/default.aspx; https://data.cnki.net/yearbook. The number of international travel agencies and star hotels reference data: the China Tourism Statistics Yearbook, https://data.cnki.net/yearbook. Meteorological data: China's meteorological data http://data.cma.cn/ (all accessed on 16 July 2020). The shortest road distance data between cities: Baidu map query (query time is July 2020).

**Conflicts of Interest:** The authors declare no conflict of interest.

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
