# Peer review of "Cultural Tourism Weakens Seasonality: Empirical Analysis of Chinese Tourism Cities"

_land, doi:10.3390/land11020308_

Round 1
Reviewer 1 Report
Style of writing.
It is not clear in the text What `SARS´ means. In the same way, please explain what is the `leave system´ in line 355.
Some expressions are repeated, this is the case of “Seasonal variation index”, which is repeated in page 3 (Line 119 and 120).
Moreover, in line 453 the Word “investigate” should be erased.
Finally, it would be advisable to avoid the schema effect, in expressions such as “Conclusion as follows:”, line 260 or “Details as follows”, line 297, or “Research shows”. Please, use more words to convey the same idea.
Literature review.
The literature review is very short. Perhaps some references to authors could be moved from the material and… section to the introduction or literature review one.
Quantitative analysis.
In “2.3.1. Variable selection” section it is said that `Foreign direct investment ´ is used as a control variable in tourism. However, it should not be considered in this analysis. There are no reasons based on tourism activities to consider this variable as a control one. It does not fit with the rest of variables considered in the quantitative analysis/model. Therefore, the model should be rebuilt excluding `Foreign direct investment ´ as a control variable. In fact, the results show that this variable is not relevant (see table 2).
Reviewer 2 Report
The studies present the situation of seasonality regarding the impact of cultural tourism in 28 typical Chinese tourist cities, in the period from 2001 to 2012.
The work is original, very complex and well written. The literature review is rich with relevant references to the topics covered.
The structure of the paper contains all necessary chapters for one scientific article.
Gini index, Tobit regression model, suitable for such research, were used for data processing.
The research results are clear and succinctly presented in the conclusions chapter.
Notes
The term “nature tourism” would be more accurate than “natural tourism” in Abstract, row 11).
The research period is 2002 - 2012. Why do you consider such an old period important and why do you think it is worth researching? The results can be relevant today, after 10 years?
It would be necessary to explain this aspect.
Please insert the source to figure 1-4. (Recommended: authors research and edited).
Reviewer 3 Report
The article deals with the important problem of assessing the seasonality of cultural tourism. The research and analysis of research results are interesting and methodologically correct, but the form of the article requires does not meet the required standards. Here are specific comments:
1) Research gaps should be identified.
2) Research objectives, problems and hypotheses should be clearly stated.
3) Conclusion and discussion chapter is not well written. It is only a summary of the research findings. A separate Discussion chapter should be written in which the authors discuss the obtained results of their own research, make a comparative analysis with other scientific works and try to explain the similarities and differences.
4) The limitations and contributions of the article to the system of science should be pointed out.
5) Cognitive conclusions are too trivial. The lower seasonality of cultural tourism, compared to e.g. leisure tourism, is obvious and results from the essence of this kind of tourism.
Round 2
Reviewer 3 Report
The authors have corrected the article according to my suggestions. I am in favor of posting an article in Land.